# The Animal Kingdom, Agriculture··· and Seaweeds

**Melania L. Cornish [1,\*], Michéal Mac Monagail [2] and Alan T. Critchley [3]** 

[1]  Acadian Seaplants Limited, Cornwallis Park, NS B0S1A0, Canada
[2]  Arramara Teoranta, Oyster Bay, Kilkieran, Co., H91 HD86 Galway, Ireland; mmacmonagail@arramara.ie
[3]  Verschuren Centre for Sustainability in Energy and the Environment, Cape Breton University,
    Sydney, NS B1P 6L2, Canada; alan.critchley2016@gmail.com
\*  Correspondence: lcornish@acadian.ca

**Abstract:** Marine macroalgae (seaweeds), are amongst the first multicellular organisms and, as such, the precursors to land plants. By the time 'land' animals arrived on the scene, terrestrial plants were plentiful and varied, and herbivorous diets developed in concert with the food sources most commonly available. However, skip forward several hundred millennia, and with the advent of agriculture, approximately 10,000 years ago, dietary diversity began to change. Today, the world is experiencing increasingly higher rates of debilitating, non-communicable diseases—might there be a connection? This paper reviews scientific evidence for the judicious use of various seaweeds in the reduction of heat stress, enhanced immunity, improved growth performance, and methane reduction in animals. The extensive, (super) prebiotic effects of selected macroalgae will also be highlighted. Key studies conducted across the animal kingdom provide considerable support that there is an overwhelming need for the guided and wise applications of increased usage of selected seaweeds in feed, food and supplements. Particular attention will be paid to the bioactive components, and nutraceutical qualities, of various seaweeds, i.e., the brown, *Saccharina* (*Laminaria*) spp. and *Ascophyllum nodosum*, and the red alga *Chondrus crispus*. Suggestions are put forward for benefits to be derived from their further applications.

**Keywords:** macroalgae (seaweeds); feed; food; *Homo sapiens*; agriculture; health

## 1. Introduction

With the lack of lignified tissues and thereby extensive fossilized evidence, it is challenging for researchers to prove definitively that seaweeds were eaten as a crucial part of the diet by early animals, including *Homo sapiens*. However, some of the tools most commonly used by scientists in their attempts to determine the components of ancient diets include reconstructions of the biomechanics of fossilized jaws, bone and teeth isotopic data, and tooth wear patterns [1]. Isotopic analysis has also been carried out on numerous fossilized remains of early hominids, the results of which indicated that before 4 mya, most hominid diets consisted primarily of $C_3$ plants (trees, fruits, shrubs, and non-grassy herbs and forbs), akin to the diets of non-human primates. By about 3.5 mya, multiple taxa began to increasingly incorporate $C_4$ foods (primarily grasses and sedges) into their diets, although the trend and ratio varied by region [2,3]. Even this information though, is problematic in determining if early humans and other animals consumed seaweeds, as the $^{13}C$-$^{12}C$ ratio varies widely amongst macroalgal species. Maberly et al., 1992 analyzed no less than 9 species of green, 15 brown, and 22 red seaweeds collected from various places around the east coast of Scotland. Their results ranged from 8.81 to 34.74%, whereas $C_4$ plants are usually around 12% and $C_3$ plants around 28% [4,5], and this effectively discounts isotopic analysis at this time, as a tool to define early seaweed consumption.

This leaves only, so far, tooth wear potentially from sand particles, and the knowledge that a stable supply of all the essential nutrients for neonatal brain growth has a high probability of exerting

dietary influence on development. This necessitates a specific period beginning with the maternal diet prior to, during, and after a lengthy gestation, followed by months, or years of nursing. An example of some of these nutrients includes polyunsaturated fatty acids (PUFAs), particularly docosahexaenoic acid (DHA:C22:6, n-3) and arachidonic acid (AA:C20:4, n-6), and critical non-residual nutrients such as zinc, iodine, and vitamin $B_{12}$ [6]. All are available to foragers in coastal marine environments around the world, and seaweeds, which have been growing along the worlds' shorelines for eons and were present well before the animals moved from the sea onto the land.

The most significant changes in human brain development occurred over the past 2.5–2 million years [7–9]. Modern-day humans now boast ownership of a precious and complex organ that functions as the epicenter of human physical existence, intelligence, and the source of all those features that define humanity. Of utmost importance to healthy neonatal brain growth and development, is the quality of the maternal diet prior to, during, and after the lengthy gestation period typical of humans. To accommodate the nutritional necessities for enhancing brain size and the associated cognitive abilities over the evolutionary long-term, a diet containing all the nutrients is likely to be a necessity over multiple generations. Family units living and eating in coastal environments 2.5–2 million years ago would have the best chances of reaching and maintaining nutritional integrity and the associated enhanced cognitive abilities. Albeit this is not proof, and absolute proof remains elusive, if not impossible, but a logical postulation.

Ironically, there is evidence today that the human brain is now actually shrinking in size. While speculation as to the reasons and implications is varied, one of the most common suggestions is that of worsening nutrition. In reality, there are most likely many factors involved and the reasons for it are quite complicated, but over the past 20,000 years, the average volume of the human male brain has decreased from 1500 $cm^3$ to 1350 $cm^3$ [10,11]. It is not yet clear what this might mean in terms of effects, if any, on cognition and/or intelligence. Still, the brain possesses a very high energy demand, and the evolutionary trade-offs are continually adapting to new niche-specific optima aimed, ultimately, at maximizing genetic fitness utilizing the substrates available [12].

The human developmental time-line reveals that early foragers eventually became tool-makers and hunter-gatherers, and finally, as population densities increased, and there were many more mouths to feed, the first steps towards agriculture took place. The consensus among archaeologists places the advent of early agricultural practices such as the domestication of plants and animals, at approximately 10,000–12,000 years before present [13,14]. There are many theories as to the reasons some very early, but geographically distant populations, concurrently took up agriculture [15], but climate change following the last ice-age is one of the most compelling [13,16]. Generally considered a watershed moment, the adoption of agricultural practices profoundly influenced humanity, and it is typically seen as a critical step towards a better life for all. However, upon closer inspection, newly emerging techniques in paleopathology, the study of disease indicators in the remains of ancient peoples, suggest otherwise. Paleopathologists can, from ancient skeletons, calculate growth rates, determine incidences of child malnutrition, and recognize scars left on bones by anemia, tuberculosis, leprosy, and other diseases [17,18].

The transition to agriculture ultimately led to significant changes in diet after thousands of years foraging for fruit, berries, roots, wild vegetation and other edibles, and then eventually the hunting of game. It is noteworthy that the timeline related to the beginning of the agricultural revolution is centuries after the era when crucial human brain development is considered to have occurred, 2.5–2.0 mya. Hominin populations by this time consisted of larger family groups who had the cognitive capacity to communicate and cooperate with one another, to hunt, and to make rudimentary tools [19]. Indeed, *H. sapiens* have spent far longer as hunter-gatherers than as agriculturists, and the transition allowed for more permanent settlements. Still, it also led to a less diverse diet and a resultant decrease in the quality of human nutrition. As groups of hunter-gatherers switched to farming, they ultimately traded quality for quantity, and the earliest crops grown were carbohydrate-based barley, wheat, rice, and corn [17], none of which contain all of the essential amino acids or vitamins necessary for human

health and survival. These crops are still farmed extensively today [20], and they make up a significant proportion of global feed and food supplies.

The increased carbohydrate content in the human diet with the shift to agriculture resulted, amongst other things, in a significant decline in dental health [17,21,22]. The relationship between dental caries and the consumption of sugar and other carbohydrates is well known, and this relationship has been used as an indicator of the dietary reconstruction as a result of agricultural intensification [23,24]. However, in 2013, Halcrow and colleagues cautioned that the carbohydrate type might have played a role, and they suggested that rice may not be particularly cariogenic. To support this standpoint, they analyzed the degree of caries in the dentition of infants and children from eight prehistoric sites in Southeast Asia. These researchers determined that while the deciduous, or baby teeth, exhibited issues related to poor dental health, the secondary, or permanent teeth, did not follow the same pattern. They concluded that while deciduous teeth were typically more susceptible to caries, the subsequent weaning of children towards a rice-based agricultural diet actually helped to maintain better oral health, contrary to evidence from regions of the world where cereals, other than rice, are utilized more extensively. An unfortunate drawback of this research relates to the unavailability of pre-agricultural samples in that region for comparison [25]. Additional declines in health potentially influenced by the adoption and transition to agriculture include a prevalence of osteoarthritis, childhood malnutrition, iron deficiency, and reduced life expectancy [17,18].

Accompanying significant changes in diet is always a corresponding change in the population diversity of gut microbes, the organisms responsible for making various enzymes and metabolites available for nutritional utilization. Alterations in macronutrient substrates available for metabolic processing create changes in nutrient supply and composition. Recent studies established that human gut microbes play multiple roles in securing the health and vitality of their host [26,27]. Mammals are metagenomic in that they possess not only their own complement of genes but also those of all of their associated microbes [28]. The contribution of the gut microbiome to the host gene pool is estimated to be over 100 times more than that of the human genome [29], and its profound influence on health and wellness is now widely recognized. The primitive human biome developed naturally in association with a variety of microbes. In an extensive sequencing study, Moeller et al., 2016 revealed that clades of the *Bacteroidaceae* and *Bifidobacteriaceae* had been maintained exclusively within host lineages across hundreds of thousands of generations, indicating robust co-speciation, and strong vertical transmission [30]. Strains of *Bifidobacteria* and *Bacteroidetes* are now known to provide significant prebiotic benefits in mammals [31,32]. Sequence analyses also provided evidence for extensive sympatry between hosts and their colonizing microbial populations [33,34]. The co-evolution of eukaryotes and their commensal, or symbiotic microbial populations played an essential role in the health and fitness of the host then, as it does now [35,36]. A significant portion of research today continues to focus on the gut microbiome, exposing the seemingly infinite number of relationships gut microbes have with their host, whether a plant, an animal, or macroalgae.

As apex consumers, humans are dependent not only on the inherent nutritional value of foods, but we are also impacted by the components that food was exposed to as it was being produced.

In addition to essential nutrition in the form of protein, carbohydrate, fat, vitamins and minerals, biologically active compounds are also necessary for optimum health, wellness, and vitality. Human health is deeply interwoven with the fabric of terrestrial agriculture, and the whole sphere of impacts is complex and far-reaching. Without agriculture, the human populations on Earth today would only exist until the food stockpiles ran out.

However, it is necessary to look upon the broader picture as a whole. In a global situation where obesity and cardiovascular and neurological diseases are at epidemic proportions and increasing, a clear assessment of the situation is warranted, even as governments begin to recognize the high costs of obesity. It is hoped that this review will provoke some thought and consideration of tools that may be naturally available in the form of macroalgae.

## 2. Bioactive Compounds in Macroalgae

It is well known that seaweeds naturally possess a plethora of unique and beneficial bioactive compounds [37–40], but adequate research in human clinical trials remains limited. Some information, however, has been derived from animal trials, and an example of this in terms of an agricultural food crop was demonstrated by Fan and colleagues, 2011. Studies showed increased antioxidant capacity and enhanced food quality in spinach grown with applications of a seaweed extract [41]. Conventional agricultural practices have been suggested by many researchers to contribute to the production of foods that are less nutritious than organically produced crops. However, this theory remains controversial, and the science needs to be better refined. Highlights of much of this research to date, however, demonstrated that while basic nutrition does not appear to differ significantly based upon culture technique, the production of various phytochemicals and bioactive compounds is more prevalent in organically grown crops, including those receiving seaweed-based inputs [42–45].

Research on the enhancement of antimicrobial activities of essential oils by the application of seaweed extract highlights an indirect benefit to humans. The mint and sweet basil in this study are traditional global medicinal plants grown on an industrial scale, and a foliar application of two doses weekly of *Ascophyllum nodosum* extract at 5 and 7 mL L$^{-1}$ for 12 weeks enriched essential oil content and quality. *A. nodosum* treated plants were more productive and showed higher antibacterial properties than the control, thus providing higher quality oils without the negative environmental impacts of synthetic fertilizers [46]. For a more thorough review on the history, development, and extensive bioactive compounds found in seaweed extracts, relative to the many benefits they can afford the agricultural industry, please see Craigie, 2011 [47].

Animal-derived products such as meat, dairy, eggs, fish, and shellfish currently represent 43% of the total protein supply for human consumption [48], and this number is expected to grow in concert with global population increases. Efforts are in play to improve the nutritional quality of certain agricultural crops by biofortification methods, particularly for essential mineral elements often lacking in human diets, such as iron, zinc, copper, calcium, magnesium, and iodine [49]. Collectively, seaweeds typically contain all of these elements and are considered a dependable source of them. Reliance by societies on agricultural production means that basic nutritional requirements are being met, but what of the other important wellness compounds mentioned here previously?

The world's populations will always be dependent upon agriculture but if we consider the current situation where global health issues continue to rise, despite the availability of agriculturally produced foods, especially in industrialized countries, is something fundamental being overlooked? It is possible, perhaps even likely, that primitive hunter-gatherers enjoyed better health and wellness than people today because they foraged for a wide variety of wild foods. Coastal diets would have been optimal, enriched in phytonutrients and their associated bioactivities, including prebiotic effects. It is challenging to determine the wellness activities of ancient peoples conclusively, and in an effort to compare energy expenditure of Westernized humans to hunter-gatherer ancestors, Pontzer and colleagues, 2012 examined daily energy expenditure and physical activity levels of present-day Hadza foragers. The Hadza lifestyle is similar in critical ways to those of Pleistocene ancestors in that they hunt and gather on foot with bows, small axes, and digging sticks, and such isolated populations for study are scarce. Over 95% of the calories in the Hadza diet come from wild foods, such as tubers, berries, small and large game, baobab fruit, and honey. From this research study, the authors concluded that Hadza individuals had lower percentages of body fat than Westerners, but contrary to expectations, total energy expenditures were similar across populations. These results add to the view that energy intake is more influential than energy expenditure in relation to obesity. Highly processed, energy-dense but nutrient-poor foods are cited as the likely culprits contributing to the obesogenic effects experienced by westernized populations [50].

Furthermore, paleopathology studies provide some evidence that life expectancy at birth in the pre-agricultural community was approximately 26 years, whereas in the early post-agricultural community, it was reduced to nineteen years [17]. Wells and Stock, 2020 developed a conceptual

framework based on evolutionary life history studies, and they applied it to better the understanding of how human biology changed in ancestral populations in association with the origins of agriculture. Their theory is based upon the assumption that energy in the form of food availability is finite and must be allocated in competition among the functions of maintenance, growth, reproduction, and defence. They argued that the origins of agriculture provoked trends in many components of biology, such as body size, fertility, and health status through the shifting of various trade-offs to new niche-specific optima [51]. Life history theory considers how organisms maximize their genetic fitness through harvesting resources from the environment and investing them in a suite of biological functions throughout their life-course [13].

A notable example of a high cost, high benefit trait in terms of a defense function is the inflammatory response. It is of high benefit because it can be life-saving during exposure to noxious challenges. Still, the high cost comes with the propensity of inflammatory defenses to interfere with normal functions. At the extreme, it can cause tissue damage, and even death (in severe cases of auto-immune disease). The inflammatory response is particularly sensitive to changes in relevant environmental factors such as an altered exposure to commensal and pathogenic microorganisms, changes in diet, antibiotics, stress, environmental and endogenous toxins, and physical activities. Chronic inflammation is universally associated with metabolic syndrome factors such as obesity [52], cardiovascular diseases [53], as well as neurodegenerative disorders [54], and cancer [55]. There have been many significant environmental changes in the past century, and it is proposed that the cost-benefit trade-off of the inflammatory response in modern human populations is not optimized to the current environmental situation, of which diet plays a primary role [56]. The prevalence of inflammatory diseases has increased significantly over recent decades, and the anti-inflammatory effects of seaweeds are well documented [57–60]

Efforts to improve the nutritional value of food crops typically focus on biofortification methods by the application of inorganic fertilizers, or specialized plant-breeding techniques, and the development of transgenic plants. It is also recognized that increasing the concentrations of bioactive substances in foods, such as β-carotene, cysteine-rich polypeptides, and certain organic and amino acids in foods helps to improve the bioavailability of certain nutrients [49]. Furthermore, farm animal production, including aquaculture, ruminant and monogastric livestock, is expected to increase by 70% to feed the anticipated human population increase to 9.6 billion individuals by 2050 [61]. In a comprehensive review, Garcia-Vaquero, 2019 presented an informative collection of studies which demonstrated numerous health and fitness benefits from the inclusion of seaweeds in the animal diet. Most of the studies are related to benefits to test-animals, fish, poultry, or shellfish, and are centered around improved growth as a function of the protein content of the macroalgae. Some of the beneficial responses also included pathogen resistance, enhanced immunity, and increased carotenoid content, suggesting more factors are at play beyond protein content [62].

Studies on the impact of foods on neurological health are advancing scientists' understanding of the dynamic interactions of the food-brain axis, and these studies have demonstrated that selected seaweeds contain compounds that are neuroprotective [63–66]. However, it is currently uncertain if, as secondary consumers, humans would receive the same neuroprotective benefits from agricultural crops (domesticated plants and animals), grown with the assistance of these beneficial seaweeds. Fan et al., 2011 showed that an extract from the brown seaweed, *Ascophyllum nodosum* enhanced the antioxidant content in spinach, which in turn, when prepared as a feed, protected the nematode, *Caenorhabditis elegans* against oxidative and thermal stress [41]. Furthermore, cows fed on a diet formulation containing *A. nodosum*, produced milk with increased iodine content and their gut microbial populations were altered in favour of increased beneficial microbes (*Firmicutes*), and a decrease in the number of *Proteobacteria* [67]. Laying hens with a 10% portion of the seaweed *Macrocystis pyrifera* added to their diet increased the n-3 fatty acid content of their eggs beyond that provided by sardine oil, improving lipid composition and consumer acceptance [68]. In addition, it is commonly understood that cold-water fish and shellfish that feed on algae are a reliable food source for the important polyunsaturated fatty acids, arachidonic (AA:C20:4, n-6) and docosahexaenoic acids (DHA:C22:6, n-3),

critical in human health and development. Humans must obtain these fatty acids from their diet, as do the fish and shellfish, by eating the (micro or macro) algae that contain them. These are examples of nutritional benefits becoming available as they advance "up" trophic levels of the food chain from primary consumer to secondary consumer and as a result of utilizing micro/macroalgal components.

It seems evident that by improving the diversity of the foods we eat beyond what current agricultural practices offer, human health and wellness status should rebound given sufficient time. Opportunities to add robust, but currently unconventional sources such as seaweeds, to animal feeds and agricultural crops, are numerous and realistic. There is a plethora of research to support the addition of specific seaweeds, or seaweed components to domesticated livestock feeds. This would potentially lead to a healthier, nutritionally diverse food supply, which will, in many cases, ultimately carry over to the end consumer. Nutrition must reach beyond the basics of protein, fat, carbohydrate, minerals, and vitamins, to include all the other substrates that fundamentally support and facilitate an optimized state of health and wellness. There are numerous, excellent scientific reviews published on the many applications and beneficial fitness effects of administering macroalgal components to livestock feed, [69–71] as examples. There is, as well, equally, or even more robust literature highlighting nutraceutical, pharmaceutical and therapeutic benefits to humans, as a search of the internet indicates.

A wide range of studies examining the bioactive properties of seaweed supplementation in domestic livestock feed are establishing species-specific, targeted applications or effects. These constituents are referred to as 'nutricines', highlighting their benefits beyond core nutrition [72]. A sample of these will be highlighted here, beginning with a study by Wan and colleagues, (2016) who carried out a 14-week feeding trial on Atlantic salmon smolts. The red seaweed, *Palmaria palmata*, was collected in winter, washed, dried, milled, prepared and administered at three different rates, 5, 10, and 15%, by weight, and made into pellets with the remaining feed ingredients. The researchers screened for physiological changes in basic haematology, immunological indicators, hepatic markers, and whole salmon body proximate composition. Results, in general, were unremarkable, as compared to controls, with the exception of a significant decrease in alanine transaminase at the 5 and 15% inclusion rates, which is thought to be a positive indicator of liver health. In addition, at the 5% level of seaweed inclusion in the salmon feed, lipid content increased compared to control fish, and an increase in lipid concentration can reflect positively on the organoleptic characteristics of the product in addition to enhanced nutrition [73].

While the purists of the world claim that wild-caught fish are nutritionally superior to farmed fish as a competent and reliable food supply, widespread systems of aquacultured fish are essential to help feed the masses in a safe and ecologically sound manner. This will be especially true in the future, and as such an important food source, fish need to be farmed carefully and sustainably. Efforts must be made to minimize stresses and promote nutritional balance in aquaculture operations. Much attention has been placed on enhancing natural immune function in farmed fish, which also improves health and growth, reduces mortality, prevents some diseases, and increases resistance to parasites. Seaweeds are amongst the most promising immunostimulants tested to date, some shown to promote growth, stimulate appetite, enhance tonicity, improve immunity, and exhibit anti-pathogenic properties [74].

Pork is the world's most consumed meat from terrestrial animals, and global commercial pork production in 2020 was estimated, by a web-based statistics company, to be around 88.73 billion metric tons [75]. It would be wise to ensure that if such a significant amount of pork is being consumed, then that meat should not only be a source of protein, fat, and specific vitamins, but should also contribute important phytocompounds. This is a potentially beneficial way to contribute to global health. However, the research is far from complete, although reported here are some examples utilizing brown seaweeds or brown seaweed extracts for livestock.

The polysaccharides laminarin and fucoidan are present in brown seaweeds, and these compounds provide important anti-inflammatory and prebiotic effects in livestock. Post-weaning pigs were fed for two weeks a diet supplemented with a 300 mg/kg laminarin-rich extract sourced from BioAtlantis

Ltd. (Clash Industrial Estate, Tralee, Co. Kerry, Ireland). The pigs had higher feed intake, growth rate, and body weight, as compared to controls. Laminarin is found in kelp-like seaweeds and is especially abundant in *Saccharina/Laminaria* spp. Corresponding to the improved fitness of the treated pigs was a measured proliferation of bacterial taxa. These included *Prevotella* that favourably enhanced nutrient digestion whilst reducing the load of potentially pathogenic bacterial taxa, including *Enterobacteriaceae* [76]. Laminarin is an important bioactive compound, and it is a robust source of antioxidants as well as an algal source of β-glucan, a natural compound known for its purported functionality in foods and its immunity-enhancing properties. However, the exact mechanisms are not yet fully understood [77]. When researchers supplemented weanling pig diets with 2.5% powdered *Laminaria* as a component of the basal feed formulation, they found that the seaweed additive not only served a nutritional purpose, but that it also exerted additional bioactivities with a positive impact on productivity [78].

Dried and ground stalks (stipes), of the brown seaweed *Undaria pinnatifida,* when fed to pigs at a rate of 1%, altered their intestinal microflora preferentially in favour of probiotic populations, such as *Lactobacillus* spp. for example, and were inhibitory to pathogens such as *Escherichia coli.* In addition, immunomodulatory effects, demonstrated by a significantly higher percentage of peripheral, blood natural-killer-cells in the treatment groups, is a promising step towards the reduction of widespread antibiotic usage [79].

Brown seaweeds, in general, are an excellent source of bioactive compounds. However, a point to bear in mind is that there is an inherent variability within and amongst species, and samples must be well characterized. In addition, there is a temptation to consider that if a little bit of something is good, then a lot should be better, and care must be taken to ensure the science is complete and thorough. Some studies showed that when certain seaweeds were used as a feed replacer, and therefore administered in higher doses, negative results such as scours and loss of conditioning in farmed animals occurred [72]. This possibility has led researchers to refine their approach and consider using seaweeds for their potent prebiotic effects as lower dose feed supplements also aimed at the potential synergies associated with whole, usually granulated seaweeds. For example, in a detailed review highlighting the benefits of macroalgae in poultry feed, it was reported that the addition of 0.5 kg of *A. nodosum* per metric ton of feed significantly reduced the effects of prolonged heat stress on the birds [80]. Certain seaweed extracts such as fucoidan and laminarin, which can focus on specific activities, are also of interest for targeted applications such as enhanced immunity and improved gut health [32,81]. Still, in consideration of poultry, Kulshreshtha and colleagues, investigated in 2020 seaweed components as agents against the drug-resistant pathogen *Salmonella Enteritidis*, carrying out the study in cell cultures. They investigated the effects of water extracts of two cultivated red seaweeds, *Chondrus crispus*, and *Sarcodiotheca gaudichaudii* in various combinations with industry-standard antibiotics. Streptomycin exhibited higher antimicrobial activity against S. Enteritidis compared to tetracycline, with a $MIC_{25}$ and $MIC_{50}$ of 1.00 and 1.63 µg/mL, respectively. However, the addition of a water extract of *C. crispus* at a concentration of 200 µg/mL to the tetracycline treatment significantly enhanced antibacterial activity (log CFU/mL 4.7 and 4.5 at $MIC_{25}$ and $MIC_{50}$, respectively). Furthermore, the *S. gaudichaudii* water extract, at 400 and 800 mg/mL, and also in combinations with tetracycline, showed total inhibition of bacterial growth [82]. The reduction of antibiotics in global agricultural operations is a very important step towards minimizing resistance to synthetic drugs, and overall, by extension, will help lead to the development of a healthier planet for people, plants, and animals naturally.

Researchers continue to explore the benefits of supplementing livestock diets with seaweeds, and whilst protein substitution is one of the more popular reasons for developing alternative feedstuffs, other vital applications are coming to light with respect to the meat and dairy industries. A sun-dried, specially managed granular extract of the fucoid, *A. nodosum* (Tasco™, Acadian Seaplants Ltd., Dartmouth, NS, Canada) was fed at a 1, 3, or 5% rate to young rams for 21 days. There was no effect under the conditions of this study on rumen fermentation, but rumen total bacteria and archaea were linearly reduced, and protozoa were linearly increased by increasing levels of Tasco™. Furthermore, the

addition of seaweed to feed reduced the total *E. coli* population, a common and ubiquitous, foodborne pathogen [83]. Other examples of the positive impact seaweeds can induce include benefits to cattle, and steers fed 20 g Tasco™/kg diet for seven days showed similar effects in pathogen reduction. Fecal shedding of *E. coli* O157:H7 was significantly reduced in both duration and intensity, indicating an inhibitory effect on the growth and proliferation of this virulent bacteria [84]. This is a valid example of a source of contamination coming from animals and negatively impacting humans, as frequent meat recalls for *E. coli* can attest.

Addition of powdered *A. nodosum* (80.0 g/cow) to feed for Holstein dairy cows increased blood glucose levels, and it decreased blood sorbitol dehydrogenase (SDH) levels. The activity of SDH in the blood of healthy animals is low, whereas its elevation above normal range implies hepato-cellular injury. The authors suggested that this result may indicate a hepatoprotective effect of the seaweed, in concert with improved energy utilization [85].

An unfortunate reality of cattle production is evidence that in the process of digesting and utilizing their feed components, bacterial fermentation in their gut releases significant amounts of methane, a particularly undesirable greenhouse gas. Of all the reported livestock produced in the world, cattle contribute approximately 62% of global emissions within the animal sector [86], and efforts to find ways to reduce this inherent methane production have become a priority. Dietary seaweeds have been shown repeatedly to influence the gut microbiome, and this holds true as well for the methane-producing microbes, including members of the Archaea. However, there remains extensive variability in effectiveness, based upon seaweed species and inclusion rates [87]. Although the rumen microbiome can ferment seaweeds and provide energy to the host animal, high variability of digestibility values is evident among and within seaweed species, and this applies to the methane-reduction effects as well [88]. Much more research is currently required, but one of the most promising seaweeds to date, for the promotion of anti-methanogenesis activity in cattle, is the red algae *Asparagopsis taxiformis,* which provided over 90% methane reduction, at a supplement level of 2% organic matter in in vitro trials [89,90]. Developing safe and effective methods for the reduction of enteric methanogenesis is indeed, challenging, and any adoptive strategies need to be sustainable, practical, and economically viable. Whilst methane reduction is not a direct nutritional benefit for humans, it does have far-reaching and important global climatic implications, and further studies on a wide range of seaweed-based dietary supplements should be undertaken.

The red seaweed *Chondrus crispus* has a long history of usage as food and medicine [91], and this species is remarkable in its bioactive characteristics, as demonstrated by various animal studies. Components of this seaweed were shown to enhance host immunity, suppressing the expression of quorum sensing and the virulence factors of a *Pseudomonas aeruginosa* strain, and enrich probiotic levels in the host [92,93]. As a supplemental feed ingredient, 2% *C. crispus* significantly increased the beneficial (probiotic) bacteria in the guts of layer hens, and it also enriched the short-chain fatty acid concentration, which is thought to act as an energy source for intestinal epithelial cells, stimulating cell growth [94]. Sulphated polysaccharides (SP) were extracted from samples of *C. crispus* collected off the Atlantic coast of Ireland and used to determine the effects, if any, on wild mussels. Results indicated that the SP from *C. crispus* rapidly induced health-enhancing activities in *Mytilus* spp. at a cellular, humoral and molecular level, and with up to a 10-day prolonged effect [95].

While scientific research regarding the impact of seaweed on equine health is somewhat limited, there is an important accumulation of anecdotal evidence for the utilization of microelements, conditioning, and other benefits derived from seaweeds. Interestingly, excessive obesity affects approximately 45% of the worldwide horse population, resulting in equine metabolic syndrome (EMS) [96], which parallels metabolic syndrome (MetS) in the human population. The same fundamentals of nutrition apply to all, and equine also utilize basic substrates to optimize genetic fitness. Still, the benefits for horses of nutricines from seaweeds have yet to be thoroughly investigated. There are several macroalgal-based products currently in the marketplace, primarily as supplements, and quality testing must be done for these to meet the appropriate regulatory criteria [97]. Horses are

not a widely used food animal for humans, but they constitute a significant proportion of agricultural livestock, and they make a large contribution to feed consumption statistics. Their general health status is but another example of the impact domestication has had on the world's livestock.

## 3. Conclusions—One IS what One Eats

It is obviously unreasonable to consider that all the members of the global, human population may return to a lifestyle of foraging, hunting, and fishing, for the acquisition of a nutritionally balanced, wild-based diet. From archeological studies, including paleopathology, it is now understood that many of the *H. sapiens* on the planet, prior to the advent of agriculture, in fact, had a more diverse diet, and are considered to have had healthier lives. In addition, as mentioned previously, the ancestors of today's humans must have had long-term access to all the essential nutrients for the growth and development of the brain with respect to its structure and sophistication—characteristics that differentiate humans from other primates [6]. Simply as a function of accessing natural food sources, useful bioactive compounds, such as antioxidants, various polyphenols, and also in the case of seaweeds, sulfated polysaccharides would be present and plentiful.

In a well-cited commentary (~2500 citations on 20 June, 2020) published in 2005, the authors remarked, "the evolutionary collision of our ancient genome with the nutritional qualities of recently introduced foods may underlie many of the chronic diseases of Western civilizations", and they provide several reasons for that statement. In particular, lower nutritional diversity, extensive food processing, and the over-consumption of sugar and salt are cited as primary factors negatively impacting the health status of global populations [98]. Without a doubt, the industrialization of the food system has further reduced dietary variability. As an example, the genetic diversity of the corn plant has now been lost, but the number of products made from this crop is in the thousands [99]. Would growing that corn with seaweed extract lead to a broader range of antioxidants for those who consume it? How many steps in the food web would maintain that bioactive nutrition? What measures must be taken to improve global health? Big questions, with complicated answers, even without addressing the various economic scenarios and their influence. It is a fact, however, that chronic diseases are impacting populations across the world, and they continue to rise, disrupting such vital organs as hearts and brains, and contributing to diabetes, obesity, inflammatory disorders, and loss of vitality. Furthermore, many of these conditions are interconnected, where one negatively influences the other, and now studies reveal that there is a strong relationship with the gut microbiome, which has co-evolved along with humans, other animals, and plants. Host dietary intake is a major environmental factor influencing gut bacterial abundances and disease phenotypes [100], and obviously, the phytonutrient quality of our food matters—especially over the longer term.

The solution, therefore, must lie within our current systems of food production, and the quality and diversity of the human diet must be improved upon. It seems quite possible that by adding seaweeds, or seaweed supplements to animal feeds, that at least some beneficial compounds will carry over to the end consumer. However, just as there are a plethora of bioactive metabolites in seaweeds, there is also a plethora of variables that must be identified and managed, and this will require significantly more research. Species differences, dosages, economics and raw material supplies must all be considered. Bioavailability of active compounds may also be a challenge as scientists strive to understand how functional constituents interact metabolically within their host [101], and potentially there will also be some individual differences in terms of response. As an aquatic natural resource, macroalgae are influenced by the quality of the environment in which they grow, and therefore it is important to be cognizant of any potential toxins or antinutrients, such as excess heavy metals.

There is no escaping the fact that with respect to food, one is, what one eats. Efforts to enrich our agricultural products by supplementing with seaweed, which contains a wide variety of phytonutrients (perhaps, 'phyconutrients' in this case) are a realistic and practical way to begin diversifying diets. There is some concern that the rates of whole dried seaweed applied may not be enough to cause a desired effect, whereas too much may have palatability or other issues. Step-wise approaches to

significant or long-term dietary changes may enable rumen microbes in cattle, for example, to adapt to higher doses over time if necessary, to obtain optimized nutritional value [70]. As an ancient food source, available thousands of years before terrestrial plants, it would seem prudent to increase the agricultural use of seaweeds as feed and food, with the aim of improving global human health. Obviously, there is some distance to go to reach that point. Still, competent and dedicated researchers are working on this opportunity, and with the eventual implementation of seaweed-derived foods and feeds into agriculture, the next 10,000 years will result in a much healthier planet, including the beings living here.

**Author Contributions:** M.L.C. and A.T.C.; formal analysis, M.L.C.; investigation, M.L.C.; writing—original draft preparation, M.L.C. and A.T.C.; writing—review and editing, M.M.M.; visualization, M.L.C. and A.T.C.; supervision, A.T.C. All authors have read and agreed to the published version of the manuscript.

**Funding:** This research received no external funding.

**Acknowledgments:** The authors are most grateful to the anonymous reviewers for their valuable and constructive comments, and also to those dedicated phycologists whom so diligently research seaweeds, with a conscientious goal to positively impact detrimental global issues.

**Conflicts of Interest:** The authors declare no conflict of interest.

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
