# Peer review of "The Animal Kingdom, Agriculture⋯ and Seaweeds"

_jmse, doi:10.3390/jmse8080574_

Round 1
Reviewer 1 Report
As mentioned by the authors, the article is a provocative review of the positive aspects that marine algae (macro and micro) can bring to the diet either directly or as a food supplement. Cornish et al, provide interesting correlations between the effects of plant domestication on the diet of Homo sapiens, aspects related to fitness, brain development, dietary diversity and chronic diseases.
Their conclusions invite to discussion on positive aspects and also strongly suggest the inclusion of algae, especially macroalgae, in the diet. It would have been interesting to contrast some of the fitness indicators that the authors indicate by decreasing dietary diversity in coastal and inland populations, with anancestral history of algae inclusion in the diet.
Author Response
Thank you very much for your thoughtful comments on this animal feed review. There really is a dearth of information relative to what early Homo sapiens were likely to have eaten both before and soon after the advent of agriculture...and how it may relate to various fitness indicators. This author did make an attempt to highlight that an investigation of that sort (in a general sense) was carried out by researchers comparing the the health status of a remote Hadza tribe to a westernized population, although dietary seaweed was not an aspect of that research.
There are very few groups such as the Hadza that have not been influenced by industrialized agriculture, and consequently, we are reliant on empirical data and information for the most part to reach our conclusions. However, information on fitness markers is readily available through paleopathology processes, but defining specific diets is a bit more challenging. I am attaching the manuscript complete with track changes for the interest and information of the reviewers. The Hadza content starts at line 190.

Reviewer 2 Report
In the attached file

Author Response
The authors are most grateful to this reviewer for such clear and concise comments regarding the content of the manuscript. Every single point was important and relevant to adjust....perhaps a testament to the reviewer's experience and keen eye. Thank you.
Attached is a PDF of the edits/revisions suggested by the reviewer with my comments regarding how each was addressed. In addition, I am also attaching a copy of the revised manuscript with track changes in place so that reviewers may see how the changes and adjustments were dealt with.
Again, many thanks to this reviewer for helping to make this is better review paper overall.
It appears I can only attach 1 document here, so I have included the PDF file with my comments as to specific adjustments/revisions made.
